# Application of ^1^H Nuclear Magnetic Resonance Spectroscopy as Spirit Drinks Screener for Quality and Authenticity Control

**DOI:** 10.3390/foods9101355

**Published:** 2020-09-24

**Authors:** Jan C. Teipel, Thomas Hausler, Katharina Sommerfeld, Andreas Scharinger, Stephan G. Walch, Dirk W. Lachenmeier, Thomas Kuballa

**Affiliations:** Chemisches und Veterinäruntersuchungsamt (CVUA) Karlsruhe, Weissenburger Straße 3, 76187 Karlsruhe, Germany; jan.teipel@cvuaka.bwl.de (J.C.T.); thomas.hausler@web.de (T.H.); katharina.sommerfeld@cvuaka.bwl.de (K.S.); andreas.scharinger@cvuaka.bwl.de (A.S.); stephan.walch@cvuaka.bwl.de (S.G.W.); thomas.kuballa@cvuaka.bwl.de (T.K.)

**Keywords:** NMR, alcoholic beverages, ethanol, methanol, acetaldehyde, screening, validation, food control, PULCON

## Abstract

Due to legal regulations, the rise of globalised (online) commerce and the need for public health protection, the analysis of spirit drinks (alcoholic beverages >15% vol) is a task with growing importance for governmental and commercial laboratories. In this article a newly developed method using nuclear magnetic resonance (NMR) spectroscopy for the simultaneous determination of 15 substances relevant to assessing the quality and authenticity of spirit drinks is described. The new method starts with a simple and rapid sample preparation and does not need an internal standard. For each sample, a group of ^1^H-NMR spectra is recorded, among them a two-dimensional spectrum for analyte identification and one-dimensional spectra with suppression of solvent signals for quantification. Using the Pulse Length Based Concentration Determination (PULCON) method, concentrations are calculated from curve fits of the characteristic signals for each analyte. The optimisation of the spectra, their evaluation and the transfer of the results are done fully automatically. Glucose, fructose, sucrose, acetic acid, citric acid, formic acid, ethyl acetate, ethyl lactate, acetaldehyde, methanol, n-propanol, isobutanol, isopentanol, 2-phenylethanol and 5-(hydroxymethyl)furfural (HMF) can be quantified with an overall accuracy better than 8%. This new NMR-based targeted quantification method enables the simultaneous and efficient quantification of relevant spirit drinks ingredients in their typical concentration ranges in one process with good accuracy. It has proven to be a reliable method for all kinds of spirit drinks in routine food control.

## 1. Introduction

The analysis of spirit drinks, which are defined in the European Union (EU) as alcoholic beverages based on some form of distillate and exceeding an alcoholic strength of 15% vol [1], is an important task for governmental food control laboratories as well as for spirit drinks producers and for commercial contract laboratories working for all stages of trade. The control of spirit drinks encompasses the control of compliance with food laws e.g., food declaration laws regarding alcoholic strength labelling [2] and specific spirit drinks laws regarding volatile and non-volatile composition [1]. An overview on the legal demands for spirit drinks is provided by Lachenmeier et al. [3]. The investigation of spirit drinks becomes more and more important for public health protection due to large-scale poisoning outbreaks due to counterfeiting and admixture, specifically with methanol [4,5]

For conventional spirit drinks analysis, several different methods have to be applied. The EU reference procedures for determination of alcoholic strength, are distillation followed by pycnometry, electronic densimetry or electrostatic balance. The reference procedure for volatile composition (especially of interest for fruit spirits and including methanol) is based on gas chromatography (GC) with flame ionization detection, and the determination of non-volatile composition (mainly sugars, e.g., found in liqueurs and compounds due to wood cask storage, e.g., found in whisky or brandies) is based on liquid chromatography [6]. Some laboratories also apply enzymatic analysis for sugars and some other compounds [7,8]. The important contaminant ethyl carbamate, found in stone fruit spirit drinks and cachaça, is typically analysed with GC combined with mass spectrometry [9]. Special analyses for some specific spirit drinks types, such as anethol in raki or ouzo or thujone in absinthe, may require even further assays [7,8,10,11].

Typically, even for a small spirit drinks analysis, at least three different methods are applied. Even the mentioned small selection of potential methods for spirit drinks analysis indicates that such an analysis is time-consuming and expensive too. Several efforts have been undertaken in the past to find a suitable screening technology combining the highest possible number of analytes into a single and preferably quick assay (screening analysis). For a long time, the screening method of choice has been based on infrared spectroscopy (IR) technologies, which can determine typically up to 10 analytes in a single assay (such as ethanol, methanol, propanol and total sugar content) [12,13]. The restriction of IR is its low sensitivity and signal overlap, making a direct analysis impossible, necessitating indirect calibration approaches based on partial least squares regression models, combining reference analysis data with a huge number of spectral data. The sensitivity of IR was also too low to detect analytes such as acetaldehyde or 5-(hydroxymethyl)furfural (HMF), typically occurring at lower ppm concentrations. 

In the last two decades nuclear magnetic resonance spectroscopy (NMR) has been proven as a primary reference method for quantitative measurement [14,15] and has been successfully introduced as a screening technology for food analysis [16]. NMR-screeners for wine [17], fruit juice [18], olive oil and honey are commercially available. As there is no NMR-screener for spirit drinks available so far, the idea of this study was to develop such a spirit drinks screener, combining as many quantitative parameters as feasible in one assay. See Table 1 for a list of validated analytes. This screener should be able to achieve insights into the quantitative composition of spirit drinks, with little manual intervention, meaning that spectra processing and evaluation need to be fully automatic, similar to the screening method already developed for alcohol-free beverages [19]. The NMR sample preparation and measurement protocol is based on previous research [20,21]. This study only deals with quantitative analysis, while non-targeted analysis, e.g., for brand assignment, has been described elsewhere [21]. 

## 2. Materials and Methods 

### 2.1. Reagents, Standards and Samples 

All Reagents and standard compounds were of analytical or high performance liquid chromatography (HPLC) grade (≥99%). Acetic acid, acetaldehyde, ethanol, formic acid, d-glucose, mannitol and methanol were obtained from Merck, Darmstadt, Germany; HMF was provided by Acros, Geel, Belgium; succinic acid was sourced from VWR, Darmstadt, Germany; isobutanol, isopentanol and 2-phenylethanol were acquired from Alfa Aesar (ThermoFisher, Karlsruhe, Germany); citric acid, ethyl acetate, ethyl lactate, fructose, 1-propanol, sodium benzoate, sodium propionate, sucrose, and 3-(trimethylsilyl)propionic-2,2,3,3-d4 acid sodium salt (“TSP”, 98 atom% D) and the reagents for the buffer potassium dihydrogen phosphate and potassium hydroxide and sodium azide were bought from Sigma-Aldrich, Steinheim, Germany; deuterium oxide (D_2_O, 99.9 atom% D) came from Deutero, Kastellaun, Germany. All aqueous solutions were made using demineralised water.

#### 2.1.1. Buffer Solution

Several buffer systems were tested during method development to ensure a uniform sample pH even from spirit drinks with different pH values without the need for further pH adjustments or titration. The optimised buffer was prepared by dissolving 20.4 g KH_2_PO_4_, 19.5 mg NaN_3_ and 100 mg TSP in approx. 90 mL D_2_O. KOH flakes were added to adjust the pH to 7.4 (at 20 °C). Finally, the solution’s volume was adjusted to exactly 100 mL with D_2_O.

#### 2.1.2. Quantification Reference Solution (“QuantRef”)

Three aqueous solutions of known concentrations were prepared from certified reference materials containing sodium benzoate (4 g/L), sodium propionate (1 g/L) and mannitol (4.5 g/L). The exact weighed portions were noted. A precise 1:1:1 mixture of these stock solutions was made, from which 900 µL was taken and mixed with 100 µL of buffer (as above) resulting in a dilution factor of 30%, compared to the initial concentrations. 600 µL of this solution was then filled into an NMR tube which was subsequently fused shut. By repeated weighing over a week the leak-tightness of the fused tube was confirmed. With this long-term stable QuantRef solution the spectrometer’s response (the ERETIC factor, see below) is determined with each new sample series.

#### 2.1.3. Quality Assurance (Control) Solution

The international standard ISO 17025 demands the check and confirmation of a standard-based quantification by an independently prepared control solution. This solution was prepared by mixing 25 mL of the 4.0 g/L sodium benzoate stock solution with 20 mL of 4.5 g/L mannitol, 15 mL of 1.0 g/L sodium propionate and 30 mL of 2.0 g/L succinic acid. Of this mixture, 900 µL was combined with 100 µL buffer (as above) and 600 µL of this finished preparation was filled into an NMR tube, fused shut and checked for leak-tightness by repeated weighing over a week. With this long-term stable monitoring solution the trueness of the spectrometer and the calculation algorithm were checked with each new sample series. 

#### 2.1.4. Spirit Drinks Matrix Samples 

To check the Spirit drinks screener’s robustness against a broad range of ingredients, five types of spirit drinks with different characteristics were used as matrix samples: Absinthe with its typical high essential oil and alcohol contentFruit spirit, having a rather high level of volatilesHerbal liqueur, with a high sugar content and a wide variety of plant extractsWhisky as a typical cask-matured spirit drinkVodka with only few minor components

### 2.2. Sample Preparation 

Before analysis, the spirit drinks were filtered or centrifuged (especially required for turbid samples). For the sample preparation, 500 µL of spirit drinks was mixed with 100 µL buffer solution, and 400 µL water–ethanol-mixture (190 mL and 50 mL) resulting in a dilution factor of 50%, compared to the spirit drink’s original concentrations. For direct measurement, 600 µL of the sample preparation was transferred into a 5 mm NMR tube (with <1% volume variation, e.g., Wilmad Labglass Inc., Vineland, NJ, USA). 

### 2.3. Proton NMR Experiments

General requirements: To ensure the correct processing of spectra, artefacts at both ends of the spectrum, resulting from the band pass filter, were distanced from the spectral region of interest (ROI) by setting the spectral width (SW) wide enough, i.e., the width of empty end regions should be > 10% of the ROI’s width, with a SW of >20 ppm a central region from −3 to 13 ppm can be processed and analysed correctly.To avoid any errors due to a non-linearity of the receiver gain (RG), all sample spectra and the spectrum of the quantitation reference (external standard) shall be acquired with the same RG setting.To ensure a near uniform excitation over the whole ROI, it is recommended to set the excitation frequency centrally in the ROI.For accurate integration or curve fitting, each signal shall be defined by a minimum of four data points in its upper half. Thus, with a typical full width at half maximum (FWHM) of 1 Hz, the resolution shall be better than 0.25 Hz

The period between pulses should be at least 5 times longer than the T1 constant of the analyte with the slowest relaxation to ensure near complete (>99%) equilibrium, otherwise the resonance signals will be attenuated and thus concentrations will be determined too low. For a defined set of acquisition parameters this attenuation is reproducible and can be compensated empirically. 

An Avance III 400 Ultrashield spectrometer, with a 5 mm selective inverse probe (SEI) with Z-gradient coils and a SampleXpress automatic sample changer (all BrukerBiospin, Rheinstetten, Germany), was used for the NMR measurements. 

All data were acquired under the control of Sample Track Client (BrukerBiospin, Rheinstetten, Germany), requiring about 35 min per sample. 

The spectral raw data were automatically processed under TopSpin version 3.2 (Bruker Biospin, Rheinstetten, Germany) to achieve the following objectives: window multiplication and Fourier-transformation, referencing the shift scale to 0.0 ppm (using the TSP-signal) and phase correction, a baseline correction (globally and selectively between 0.001 ppm to 0.97 ppm; 1.388 to 3.4 ppm; 3.9 to 4.75 ppm and 4.95 to 9.999 ppm, the exclusion zones span the resonances of ethanol and water). Finally, all processed spectral data were saved on a server. 

Three different NMR experiments were measured for each sample: A one-dimensional ^1^H-NMR with suppression of the water signal by simple presaturation, a one-dimensional ^1^H-NOESY and a two-dimensional ^1^H-^1^H J-resolved (JRES) (= coupling-resolved), both using shaped pulses to suppress the eight intensive signals of water and ethanol. 

These experiments are based on Bruker’s standard NMR experiments [17] with minor modifications, foremost, among them an additional delay after acquisition allowing for full relaxation of the spins and keeping D1, the regular relaxation delay, short. During D1 the presaturation is performed, a short D1 ensures that less energy is transmitted into the sample for a maximum of sample temperature stability. The detailed settings described below have been validated as fit for purpose. 

For each sample the first experiment was used to optimize the hard 90° pulse with Bruker’s PULSECAL routine and the power of the presaturation is calculated accordingly (and limited to 25 Hz), the optimised values were then stored for usage in the following experiments. Furthermore, the shaped pulse was optimised to suppress the water and ethanol peaks (8-fold suppression) in the following NOESY (Nuclear Overhauser Enhancement Spectroscopy) and JRES (J-resolved = coupling-resolved) experiments.
Pulse program: zgpr.mod_d7Time Domain: 64k (=65,536 data points)Dummy scans: 4Number of scans: 4Spectral width: 8.2 kHz (20.55 ppm)AQ Acquisition time: 3.985 sSFO1 (excitation/observation frequency): on resonance with the water signal D1 (pre excitation delay/ pause): 4 s (phase with presaturation, before excitation)D7 (post acquisition delay/ pause): 8 s (phase after acquisition, without presaturation)Digmod (digitalization mode): Baseopt (a Bruker function for digital filtering of the FID ensuring no 1st order phase distortion, no distortions at both ends of the spectrum, a baseline of exactly 0 if no other effects distort the FID and no attenuation of signals at the outer edges of the spectrum)TE (sample temperature): 300.0 K (±0.2 K)Size (SI): 128k (=131,072 data points)LB (line broadening): 2 HzWDW (window function/apodization): EM (exponential multiplication)

A one-dimensional ^1^H-NOESY spectrum was recorded as the second experiment, it was later used to quantify the analytes. The parameters are:
Pulse program: noesygppr1d.comp1d7Time Domain: 64k (=65,536 data points)Dummy scans: 4Number of scans: 32Spectral width: 8.2 kHz (20.55 ppm)AQ: 3.985 sSFO1: on resonance with the water signal D1: 4 s (phase with presaturation, before excitation)D7: 8 s (phase after acquisition, without presaturation)Digmod: baseoptTE: 300.0 K (±0.2 K)Size (SI): 128k (=131,072 data points)LB: 0.3 HzWDW: EM

The FID (free induction decay) of the one-dimensional-NOESY was processed with window functions: one variant by using an exponential function to enhance the signal-to-noise ratio (SNR) for quantification and the other by using a Gaussian function to improve resolution to precisely determine peak maxima. An automatic first order phase correction and a baseline correction (leaving out the signal regions of TSP, Ethanol and water) were applied to the spectra. Two peak lists were obtained from both processed spectra.

The third experiment performed, a two-dimensional JRES, was used to detect the analytes’ presence in the sample and the exact chemical shifts of the characteristic peaks. The parameters for the JRES are:
Pulse program: jresgppsqfAQ_mod (acquisition mode): DQD (digital quadrature detection)FnMODE (Acquisition mode of the indirect dimension): F1 QFTime Domain:F2 8192, F1 40Dummy scans: 16Number of scans: 8Spectral width: F2 16.70, F1 0.195AQ: F2 0.613 s, F1 0.26 sD1: 1 sDigmod: digitalTE: 300.0 K (±0.2 K)Size (SI): F2 16k, F1 256

The JRES spectra were processed in Topspin with a squared sine bell window function, tilted and symmetrised. 

### 2.4. Automated Analysis of the NMR Spectra 

The TopSpin-processed NMR spectra were analysed using Matlab (version 2015b, The Math Works, Natick, MA, USA). The general process of the Matlab script runs as follows: 

First the raw spectral data, the peak and peak intensity lists (generated by TopSpin) and the excitation pulse lengths of all spectra were imported into Matlab’s workspace. The full data sets were then cropped leaving only the data points between −3 ppm and 14 ppm. 

The presence or absence of each analyte in a spirit drink sample was determined by checking the two-dimensional JRES spectrum for the analyte’s specific peak pattern. Furthermore the JRES yielded the exact chemical shifts of the characteristic peaks in each sample. These exact shifts were subsequently used by the curve fitting algorithm. 

The one-dimensional ^1^H NMR spectra were optimised by a reference deconvolution algorithm that removed minor asymmetries (due to slight field inhomogeneities) from the signals. The integral (area under the curve, AUC) of each characteristic signal was calculated with a Matlab based curve fitting algorithm focusing on the exact chemical shifts yielded by the JRES and taking into account the known coupling constants of multiplet signals. 

The ERETIC factor (the spectrometer’s sensitivity expressed in a.u. × ppm × L/mol) was determined from five resonances of the QuantRef sample (see Table 2) measured in one series with the spirit drink samples and quantification was then performed using the ERETIC–PULCON method (see Section 2.4.4). 

#### 2.4.1. Shim Quality and Reference Deconvolution 

In each spectrum (QuantRef and samples) the TSP signal was fitted (using the method described in Section 2.4.3), the fit integrated and the FWHM calculated. If the TSP-FWHM of a sample was wider than 1.3 Hz and/or the intensity of the TSP resonance fell outside of a predefined expected range, further analysis of this sample was rejected and a warning issued to the user. Potential causes for TSP resonances above the threshold value of 1.3 Hz could be e.g., an accidentally bad shim (each sample is automatically gradient-shimmed by the spectrometer), which will deteriorate all signals of the spectrum, or an aggregation of the TSP molecules e.g., as ligands around a suitable central cation. If the TSP-signal is narrower than the defined FWHM limit, it is subsequently used to achieve a reference deconvolution using the FIDDLE algorithm (Free Induction Decay Deconvolution for Lineshape Enhancement) [23,24] on the zero-filled, Fourier-transformed and phase corrected experimental spectral data. Minor magnet field inhomogeneities will slightly deform all NMR peak shapes in a spectrum in the same way. The reference deconvolution will remove such shape errors from the whole spectrum. 

By computing the inverse Fourier transformation of (a) the experimental TSP peak’s data and (b) the optimised fit of the TSP peak (a very clean and sharp singlet) and pointwise division of (a) by (b) a correction function can be obtained. Pointwise division of the complete experimental spectrum’s FID by the correction function and Fourier transformation then yields the deconvoluted spectrum. 

Further processing of the spectra differs between the QuantRef sample and the spirit drinks samples: Since the composition of the QuantRef is exactly known and its spectrum has only a few clearly separate peaks, signal curve fitting and subsequent integration may reliably be based on known values for the chemical shifts and coupling constants of the reference substances. 

From each of the five integrals an ERETIC factor is calculated with formula 3 (see Section 2.4.4). From these five factors the arithmetic mean and standard deviations were calculated. The variation coefficient of each singular ERETIC factor must be smaller than 2%, otherwise a warning message is issued and the evaluation aborted. If all is well, the mean ERETIC factor is stored and used for all following quantifications in this sample series. 

The next step in the automated data evaluation is the calculation of the quality assurance (control) solution’s recovery rate: After checking the TPS’s FWHM to be below the threshold and a reference deconvolution, the reference substance’s signals (see Table 2) are curve-fitted and integrated to yield their PULCON-calculated concentrations. A successful comparison with their known concentrations (no recovery more than 10% off) confirms that both the QuantRef and spectrometer are in order. 

#### 2.4.2. Analyte Identification and Precise Shift Determination 

Due to the wide variety of spirit drinks compositions, it was expected that fixed values for the analytes’ chemical shifts might sometimes differ too much from the real values in individual samples where e.g., a pH- or concentration-induced signal shift could occur. 

To ensure the correct identification of an analyte in a spirit drink sample and the determination of its exact signal position(s) the JRES spectrum was used: Its advantage is the dispersion of multiplet peaks into the second dimension resolving many overlapped signals and enabling the clear determination of coupling constants. 

Common experience shows that the chemical shifts of a given analyte may vary, depending on supramolecular influences (e.g., pH, solvation and concentration), but the intramolecular scalar coupling constants have proven as reliably invariant. A JRES spectrum is basically a three-dimensional dataset in a matrix where the chemical shift (F2) changes from column to column and the couplings (F1) are resolved between rows, the central row containing the spectral data with a coupling of zero (singlets). Each matrix element contains the intensity at this point in the “spectral landscape”. If the multiplet type and the coupling constant *J* for a searched characteristic resonance and the acquisition parameters of the JRES are known, one can calculate from the resolution in F1 the rows which will contain the peaks of this multiplet (if the sample contains the analyte). E.g. in a JRES spectrum with 0.3 Hz resolution in F1 a triplet with *J* = 7.2 Hz will have intensity maxima in the central row and 24 “side” rows above and below the central row. 

A limited subset was then temporarily extracted from the full JRES dataset, containing only one of the “side” rows where a peak of the multiplet should appear and a reduced range of chemical shifts centred on the typical value for the examined resonance. Using a publicly available Matlab function “peakfinder” [25] the intensity maximum was searched, thus yielding the exact column index (i.e., chemical shift). The algorithm developed by our team then checked if the intensity of the found peak was significantly higher than the other data (noise) in the evaluated row and, since multiplets must be symmetrical to *J* = 0 Hz, if the multiplet’s complementing peaks could be found at the other expected positions in the same column of JRES data. After confirmation of the multiplet’s existence and determination of its exact chemical shift (in this sample), precise starting values for the curve fitting were calculated. 

#### 2.4.3. Curve Fitting

The signals to be evaluated were fitted with a pseudo-Voigt curve
(1)$$I_{fit}=\;\eta \;\cdot \;\frac{A}{1+{\left(\frac{x-x_{max}}{\gamma }\right)}^{2}}\;+\;\left(1-\;\eta \right)\;\cdot \;A\;\cdot \;e^{-\ln2\;\cdot \;{\left(\frac{x-x_{max}}{\sigma }\right)}^{2}}$$
where: *I*_fit_ is the fitted NMR signal intensity,*η* is the weight of the Lorentzian contribution (1 = fully Lorentz-shape, 0 = fully Gauss-shape),*A* is an amplitude factor to adapt the fit to strong and weak peaks,*x* is a variable value on the chemical shift scale,*x_max_* is the function’s center, the chemical shift of the maximum of the peak to be fitted,*γ* is the half width at half maximum of the Lorentzian contribution, *σ* is the half width at half maximum of the Gaussian contribution of the peak to be fitted. 

These pseudo-Voigt profiles were fitted to the data points using a simple least squares cost function:(2)$$C=\;\overset{x_{end}}{\underset{x_{start}}{\sum }}{\left[I_{fit}\left(x\right)-\;I_{raw}\left(x\right)\right]}^{2}$$
where: *C* is the sum of all residues (over the whole range of the fit),*x* is a variable value between the fit region’s borders on the chemical shift scale,*I_fit_*(*x*) is the fitted NMR signal intensity at *x*,*I_raw_*(*x*) is the measured NMR signal intensity at *x*.

Using a publicly available Matlab function “Fminsearchbnd” [26] the parameters *η*, *A*, *x*_max_, *γ* and *σ* were iteratively adapted to optimize the fit, arriving at a minimum of C. To focus the optimization on the relevant signal even in a region congested with irrelevant other peaks, constraints were determined and applied to the fit: Each signal chosen for evaluation has characteristics such as chemical shift, FWHM, multiplet type and (where applicable) coupling constant(s) and roof effect ratios. For these characteristics, upper and lower boundaries were set: The fit may adapt *x*_max_ ± 0.01 ppm from the chemical shift found by evaluation of the sample’s JRES spectrum (or the defined values in case of the QuantRef), the widths (FWHM) can be varied between 0.5 and 2.5 times the initial value of 1 Hz. Coupling constants used to find the maxima of a multiplet may be changed less than ±5% from the values determined from the JRES. 

To fit a multiplet of *n* peaks, we simply fitted a sum of *n* pseudo-Voigt curves with different *x*_max_ (calculated from the multiplet’s splitting) and different coefficients to *A* taking into account the Pascal’s triangle intensities pattern and possible roof effects (dependent on the spectrometer’s basic acquisition frequency and thus empirically determined). See Figure 1 and Figure 2 for examples and a visual explanation.

#### 2.4.4. Quantification 

Analyte concentrations were calculated from the intensities (integrals) of their characteristic resonances using the PULCON method (Pulse Length Based Concentration Determination) [27,28].

Basically, this well proven method uses an external standard of known concentration(s), called QuantRef (= quantification reference solution), to determine the NMR spectrometer’s response to a specific number of resonating nuclei under given acquisition conditions. This normalised response/sensitivity is called the ERETIC factor (Electronic Reference To access In-vivo Concentrations) [28]. 

The following equation was used to calculate the ERETIC factor:(3)$$f_{\mathrm{ERECTIC}}=\frac{I_{\mathrm{Ref}}\;\cdot \;SW_{\mathrm{Ref}}\;\cdot \;M_{\mathrm{Ref}}}{SI_{\mathrm{Ref}}\;\cdot \;\rho_{\mathrm{Ref}}\;\cdot \;N_{\mathrm{Ref}}^{H}\;\cdot \;f_{\mathrm{QR}}^{\mathrm{dil}}}\;$$
where: *I*_Ref_ is the absolute integral (Ref = reference) of a selected resonance of the reference substance,*SW*_Ref_ is the spectral width (20.5504 ppm),*M*_Ref_ is the reference substance’s molecular weight (144.11 g/mol for sodium benzoate, 182.17 g/mol for mannitol, and 96.06 g/mol for sodium propionate),$N_{\mathrm{Ref}}^{H}$ is the number of protons (per molecule) generating the selected resonance,$\rho_{\mathrm{Ref}}$ is the reference substance’s exact mass concentration,*SI*_Ref_ is the size of the real spectrum, which shows the number of data points after Fourier transformation (131072),$f_{\mathrm{QR}}^{\mathrm{dil}}$ is the QuantRef’s dilution factor (0.3) resulting from its preparation.


Evaluating the sodium benzoate peak between δ 7.61 and 7.525 ppm, *n*^H^ = 1, the mannitol doublets between δ 3.92 and 3.84 ppm, *n*^H^ = 2 and between δ 3.73 and 3.64 ppm, *n*^H^ = 2, the sodium propionate quartet at δ 2.25 to 2.12 ppm, *n*^H^ = 2 and its triplet at δ 1.12 to 1.00 ppm, *n*^H^ = 3 an average ERETIC factor was calculated from these signals and used for quantification. The relative deviation between the five individual ERETIC factors shall be lower than 2%. Otherwise the QuantRef solution needs to be checked and possibly be freshly prepared. 

The following PULCON equation was used to calculate the analyte concentration $\rho_{X}$ in unknown samples:(4)$$\rho_{X}=\frac{I_{X}\cdot SW_{X}\cdot M_{X}\cdot NS_{\mathrm{QR}}\cdot P1_{X}}{SI_{X}\cdot f_{\mathrm{ERETIC}}\cdot N_{X}^{H}\cdot NS_{X}\cdot f_{X}^{\mathrm{dil}}\cdot P1_{\mathrm{QR}}}$$
where: *I*_X_ is the absolute integral of the evaluated peak,*f*_X_^dil^ is the sample’s dilution factor (0.50),*NS*_X_ and *NS*_QR_ are the number of scans for the sample spectrum and number of scans for the reference spectrum respectively,*SI*_X_ is the size of real spectrum (131072),*P1*_X_ and *P1*_QR_ are the respective pulses,$f_{\mathrm{ERETIC}}$ is the ERETIC factor (see Equation (1)).

If more than one signal was used for the determination of an analyte, the Matlab script automatically checked that the standard deviation stayed under a threshold (20% for fructose, 8% for sucrose and glucose, 10% for isobutanol and 5% for all other analytes). If the values passed this test, the mean of the calculated concentrations was saved as the result; otherwise the analyte was noted as “not quantifiable”. 

The Matlab script saves the results as excel and text (txt) files. This enables the direct import in the laboratory’s LIMS system (Limsophy, AAC Infotray AG, Winterthur, Switzerland).

### 2.5. Quality Assurance 

The QuantRef’s calculated concentrations are documented in a control chart and shall not vary more than ±5% from their original values. Another internal quality control of each sample’s measurement is the FWHM of the TSP (internal standard) peak. It shall not exceed 1.3 Hz, otherwise the measurement or even the sample preparation has to be repeated. A FWHM higher than 1.3 Hz could result e.g., from a substandard shimming or a turbid sample.

For quality assurance, the last sample in each measurement series is the control solution (see above). From each new measurement the concentrations of the control solution’s substances are calculated and may not vary more than ±5% from the original values at preparation (recovery rate between 95% and 105%).

### 2.6. Validation Experiments 

Considering the lessons learned in previous validations of quantitative multicomponent NMR assays [29], this method was validated using real matrix samples (commercial spirit drinks) and not just in blank aqueous/ethanolic solutions spiked with a reference substance.

#### 2.6.1. Working Range and Measurement Uncertainty 

For each analyte, a stock solution was prepared in an ethanol–water mixture (190 mL + 50 mL, ca. 21% vol). To determine linearity, the limit of detection (LOD) and the limit of quantification (LOQ), samples at 9 different concentration levels were prepared by spiking a 35% vol ethanol–water mix with the stock solutions and measured by NMR. Appropriate concentrations were chosen regarding the concentrations typical for the selected analytes in spirit drinks or taking into account the maximum content as defined by legal food regulations (see Table 1). At each concentration level four sample preparations were done and measured to evaluate the preparation variance. 

The Measurement uncertainty was then calculated with an in-house QM-approved Excel script. Variances of the four repetitions at each concentration level were calculated and a weighting function was determined using a best-fit function. The weighted measurement results were then used to determine the upper and lower confidence intervals at each concentration point. 

To determine the relative measurement uncertainty over the working range, the uncertainties were extrapolated against the concentration 0 mg/L and the found blank value *β*(0) was used as the offset for the measurement uncertainty. The function Δ*β* = *β_found_* × 8% + *β*(0) was then used as the concentration-dependent measurement uncertainty. This function is stored in the Matlab script. Thus, Matlab will output the calculated concentration value and the associated uncertainty.

#### 2.6.2. Recovery and Matrix Effects 

To check for matrix effects, the five different spirit drink matrices (see above) were spiked with the analyte stock solutions to prepare four different concentration levels in the range typical for each analyte in each spirit drink matrix. Each of the five spirit drink matrices was also measured without spiking to yield blank values. If a matrix contained an analyte beforehand, a series of five dilutions was prepared by adding a 35% (vol) ethanol–water mix to the spirit drink matrix. 

#### 2.6.3. Specificity and Selectivity

From NMR-spectra of spirit drink matrix samples spiked with analytes the signals specific for each analyte were identified and integration regions were determined which were not overlapping with other signals. 

#### 2.6.4. Sample Stability

For laboratories with a high sample throughput the stability of a sample over time is important, because measurement does not always immediately follow sample preparation. To determine the sample stability over time, three samples of each matrix were prepared and each sample was measured four times at intervals of about one day. The stability was evaluated by the slope of the compensation line of all four measuring series.

## 3. Results 

### 3.1. Relevant Analytes and Their Characteristic NMR Signals 

Vast varieties of spirit drinks are offered for consumption, and their qualitative and quantitative compositions differ considerably. Of the 20 different spirit drinks ingredients initially identified as potential analytical targets, 15 were found to be accessible by NMR analysis; these are summarised in Table 1 and Table 3. 

For the initial evaluation whether a quantitative NMR spectroscopy approach is suitable for the potential analytical targets, proton spectra of commercially available pure compounds were acquired and compared to spectra of real (commercial) spirit drinks samples. For the majority of the tested substances, direct quantification by simple integration of their resonance signals was not possible due to insufficient separation from other signals or due to minor signal shifts depending on the analyte’s concentration or the solutions pH-value. Especially the mid-field region (around 3.3 to 4.1 ppm) of ^1^H-NMR spectra of multi component mixtures is rather crowded, the strong and often (partially) overlapping signals of carbohydrates (mostly sugars) and other substances appear there. Strong neighbouring signals of major components can occlude other smaller peaks of less concentrated minor ingredients. This is a common problem and has been described earlier, for example for the NMR-analysis of honey samples [30]. 

Attempts to evaluate JRES data for quantification were not satisfying in spite of the advantages of JRES: its increased spectral dispersion in a second dimension and the clear information about the *J*-coupling. On the other hand, the lesser sensitivity of JRES and the overlapping of the dispersive tails of nearby signals are obstacles to the quantification (especially at lower concentrations) [31]. 

Thus, we found the one-dimensional ^1^H NMR spectrum more suitable for quantification. Nevertheless, in the current context the JRES experiment has proven valuable and reliable in confirming the presence of analytes in the needed concentration range and determining the exact chemical shifts of the analytes in each sample. The information obtained from the JRES was then subsequently used for the curve fitting to analyse the one-dimensional ^1^H NMR spectrum. 

For five compounds (anethol, furfural, malic acid, 1-pentanol and isopropanol) recoveries were found to be unreliable or too low (<80%), LOD and LOQ too high, or the separation of their signals from other resonances was unreliable. Finally, for the fifteen compounds listed in Table 1 and Table 3 the test method was validated with results fit for purpose. 

In aqueous solutions the carbohydrates glucose and fructose exist in equilibrium of different anomeric and tautomeric forms (Figure 3). For glucose, the α-and β-anomers of glucopyranose predominate with 36% and 64%, respectively. At least five tautomeric forms of fructose exist in aqueous solution [32]. Furthermore, the presaturation used to suppress the water resonance slightly attenuates signals in the vicinity as well, e.g., the resonances of the anomeric protons of glucose located around 4.6 and 5.2 ppm [33]. Taking into account both the signal attenuation by the water suppression and the anomeric/tautomeric equilibria, the calculation correction factors for glucose and fructose were determined by analysing spirit drinks samples spiked with varying concentrations of glucose and fructose.

Figure 4 and Figure 5 show spectra with enlargements of the characteristic resonances used for identification and integration of the fifteen analytes. Table 3 lists the chemical shift ranges of the characteristic signals, their multiplicity, coupling constants and the number of protons (per molecule) giving rise to each signal. 

### 3.2. Analytical Limits

The analytical limits for the validated analytes, calculated complying with the German Standard DIN 32645:2008 and using a validated and QM-approved Microsoft Excel worksheet, are listed in Table 4.

The LODs varied in the range of 2 mg/L to 74 mg/L. The lowest value was attributed to methanol (2 mg/L), followed by formic, citric and acetic acids (3 mg/L) while the highest value was found for isopentanol. The LOQ values were in the range from 5–173 mg/L for methanol and isopentanol, respectively. 

Of the three sugars, fructose showed the highest LOD and LOQ values (56 and 132 mg/L). Still the LODs and LOQs of the analytes are lower than or at the low end of the relevant concentration ranges for the analysed spirit drinks ingredients. Thus, NMR spectroscopy can be considered as a tool suitable for the monitoring of spirit drinks by official food control.

### 3.3. Overall Measurement Uncertainty 

The determined uncertainty measurement of analytes in spirit drinks are summarised in Table 5. The recovery rates were set at 100%. Any deviations in recovery are already included in the determination and verification of the measurement uncertainties. Recovery errors are considered purely statistical errors and not systematic errors. A recovery rate correction is therefore not performed.
Δ*β* = *β_found_* × 8% + *β*(0)(5)

### 3.4. Stability

The general stability of the samples is judged as sufficient for two days after preparation, see Table 6 for details. The observed deviations due to ageing of the samples stay under the 8% threshold determined as the overall uncertainty. Acetaldehyde and formic acid show a loss above threshold even after one day. If there is an objection concerning these analytes, the measurement shall be confirmed using a complementary analytical method or a NMR measurement immediately after sample preparation. 

## 4. Discussion

A fit for purpose level of accuracy (the routine screening of spirit drinks) was proven for the fifteen analytes in all spirit drinks matrices except for methanol in herbal liqueur. Due to the typically low concentrations of methanol in herbal liqueur and the rather high glucose and sucrose contents, which disturb the baseline in the range of the methanol signal, too high concentrations of methanol were found. In case of a complaint about methanol in herbal liqueur, the baseline around the methanol resonance must be checked especially if there is a positive bias. 

For ethyl lactate, a minor positive bias was found in fruit spirit and absinthe. Since the deviations are only small, further validation is not deemed necessary. 

For citric acid in herbal liqueur and absinthe sometimes wrong contents were found. This is due to an extreme signal overlap by accompanying substances. For citric acid, correct values are obtained only for whisky, vodka and fruit spirit. In case of conspicuously low or high citric acid values, it is necessary to check the line shape of the citric acid signal and confirm the result with another measuring technique.

### 4.1. Comparison with Other Methods 

In 2018 our lab took part in an interlaboratory proficiency test. Table 7 shows the z-scores achieved with the NMR spirit drinks screener: 

### 4.2. Application in Routine Analysis

With rising global trade and new products on offer every year, official food and beverage control needs to stay innovative to cope with coming analytical challenges. The increasing number of samples per year (e.g., due to merging and specialization of official laboratories in Germany) furthermore necessitates accurate and efficient measurement and data processing workflows enabling high-throughput sample analyses. 

Some NMR analyser packages were already commercialised (e.g., “Juice Screener” and “Wine Screener” by Bruker [18,34]) and several current research projects are striving to deliver solutions for more analytical questions. Typically the commercial analysis packages are fully automated and installed as “black box” systems on NMR spectrometers. Users cannot enhance or adapt these test methods on their own to cope with new challenges or to analyse different matrices with such proprietary software. 

The automated procedures for the analysis of spirit drinks developed in our laboratory successfully fill the demand for an efficient simultaneous multi-analyte test method and are open for further development. Besides common commercial spirit drink samples, our lab investigated several dozens of samples of raw spirit (drinks) and “moonshine” (esp. eastern European “Samogon”). The method is already validated for several volatiles (fusel oils) relevant in raw spirits.

To start the automated data evaluation, the operator only needs to input the data path to the selected experimental series containing the spectra of samples, the QuantRef and the QA sample. The reports (in excel and txt file format) for the evaluated samples list the concentrations of the 15 analytes in each sample and can be imported into LIMS directly. A high throughput is possible, as the algorithm analyses large numbers of samples without human intervention, within a reasonable length of time (e.g., evaluation of 25 samples takes under 5 min on a Quad-Core 3 GHz CPU, 16 GB RAM computer system). 

Taking into account the spectrometer’s cost of purchase and annual operating costs, personnel costs and costs for consumables (all 2020 values), a pessimistic calculation of total costs per sample analysis adds up to approximately 50 EUR (60 USD). This amount factors in lower than 100% capacity utilization (due to research and development work, maintenance downtime etc.). For high throughput routine NMR experiments resulting in near full utilization, the costs per sample can be distinctly lower. Considering that one NMR experiment will yield quantitative data for several compounds in a sample, the application of NMR spectroscopy is comparably cheap.

## 5. Conclusions

NMR spectroscopy offers a quick, reliable and efficient methodology to quantify relevant ingredients in spirit drinks fulfilling the requirements of the standard DIN ISO 17025 “General requirements for the competence of testing and calibration laboratories” [35]. The method’s accuracy and analytical limits enable the monitoring of legal maximum limits (e.g., for methanol) as well as the controlling of manufacturer declarations. Besides, the performance parameters of the presented spirit drinks analysis by NMR spectroscopy are similar to other methods used with this aim. The integrated workflow of the automatic routine described here performs all steps necessary for quantitative analysis: spectra import, extraction of data points, curve fitting the signals of interest, integration of the optimised fit, quantification based on the PULCON principle and output of the results as report files. The method developed provides a streamlined, yet flexible data processing workflow for NMR analysis of spirit drinks, which can be simply adapted to the analysis of other matrices.

This method offers the potential to simultaneously quantify even more substances typically present in spirit drinks (or other foodstuffs): Any substance containing NMR-active nuclides, e.g., ^1^H, ^13^C, ^19^F…) will give rise to one or more NMR resonance signals. The pattern of these signals (their chemical shifts and multiplet splittings) is characteristic for the substance, especially if the substance has NMR-active nuclei in several different chemical moieties. The relative intensity of these signals is quantitatively linked to the stoichiometric number of nuclei giving rise to this resonance. NMR has been proven as a metrological primary method of measurement [14]. Thus, if the spectrometer’s sensitivity has been determined once with a certified 1H reference material, one can quantify any other proton-containing substance. However, some caveats must be considered: 

One downside of NMR is its comparably low sensitivity: Nowadays, NMR spectrometers with 7 to 14 T field strength (resulting in proton Larmor frequencies from 300 to 600 MHz) offer a good compromise between pricing, stability/reliability and a wide application range for routine quantitative analyses. At these magnet strengths, the LOD/LOQ lies in the 5 to 50 ppm mass concentration range. For small molecules (e.g., methanol) 5 mg of analyte per kilogram sample convert to 5 mg/kg/32 mg/mmol = 0.15 mmol/kg. If a longer experiment time (to accumulate more FID scans) is acceptable or a cryo probe (having a considerably lower electronic noise level) can be used or works with a magnet of higher field strength, the LODs/LOQs will be significantly lower. 

For very complex mixtures, another challenge can be the “crowding” of signals, especially in the chemical shift range around 3.3 to 4.1 ppm, where many carbohydrate resonances appear. The unambiguous assignment of signals to their molecules is not easy and often only possible using a two-dimensional spectrum, such as the JRES. If signals from different nuclei overlap, they need to be deconvoluted prior to quantification, using complex curve fitting routines, valid knowledge of coupling constants and the relative signal heights in multiplets. Thus, further development will focus on streamlining and flexibilisation of the algorithm code and on improvement of the quantification of compounds not yet adequately separated from interfering signals.

## Figures and Tables

**Figure 1 foods-09-01355-f001:**
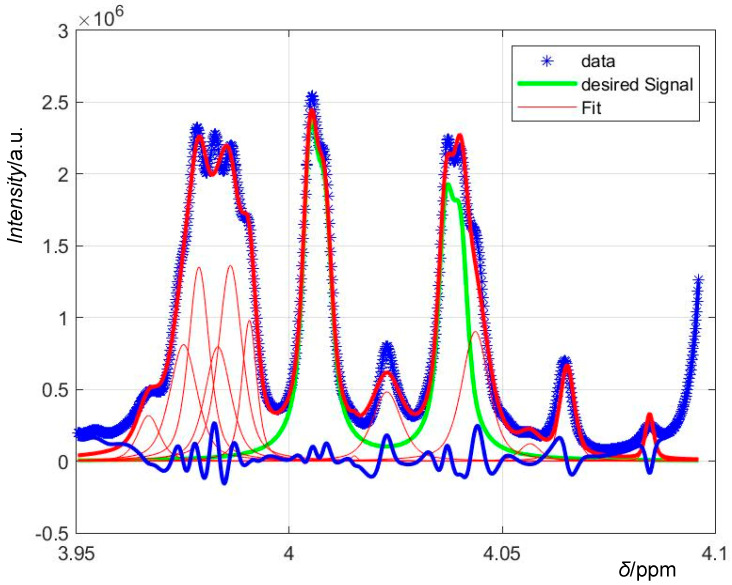
On demand Matlab output graphs of the curve fits, in this case the first signal of fructose. The green curve is the desired fit, the sum of four pseudo-Voigt profiles with the constraints typical for this signal. To mimic the curve of the original data (blue asterisks), Matlab adds more singlet profiles (thin red lines), all combined yield the thick red line. The residues are shown as a thick blue line.

**Figure 2 foods-09-01355-f002:**
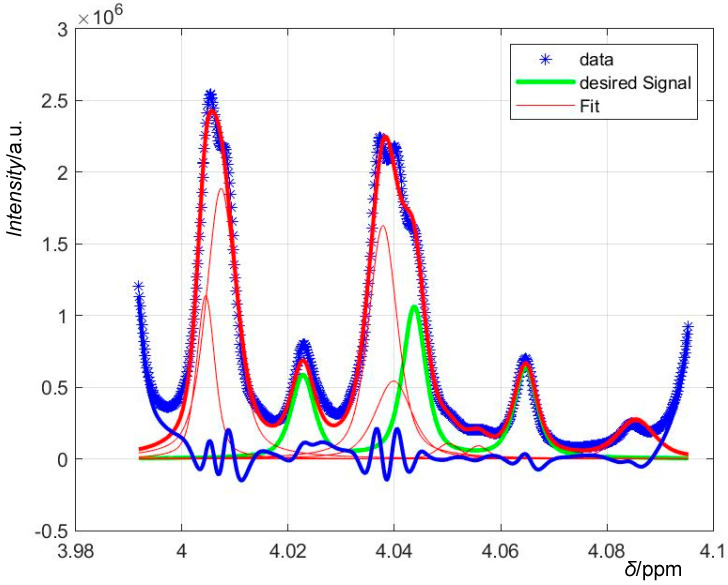
Curve fit of the sucrose triplet (thick green line). Although the resonance is not at all separated from the fructose double-doublet (compare Figure 1), the algorithm can again separate the desired signal from interfering signals (approximated as simple singlets, the thin red lines) arriving at a good representation of the original data (blue asterisks) with acceptably low residues (thick blue line).

**Figure 3 foods-09-01355-f003:**
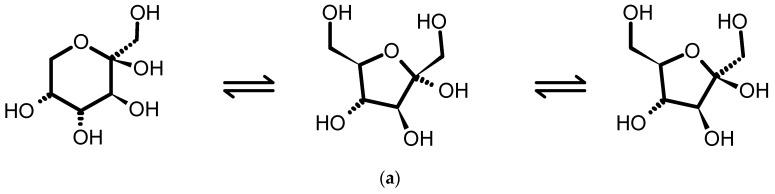
The predominant forms of fructose and glucose in aqueous solution: (**a**) fructose equilibrium 76% β-D-Pyr, 4% α-D-Fur, 20% β-D-Fur; (**b**) glucose equilibrium of 36% α-D-Pyr, 64% β-D-Pyr.

**Figure 4 foods-09-01355-f004:**
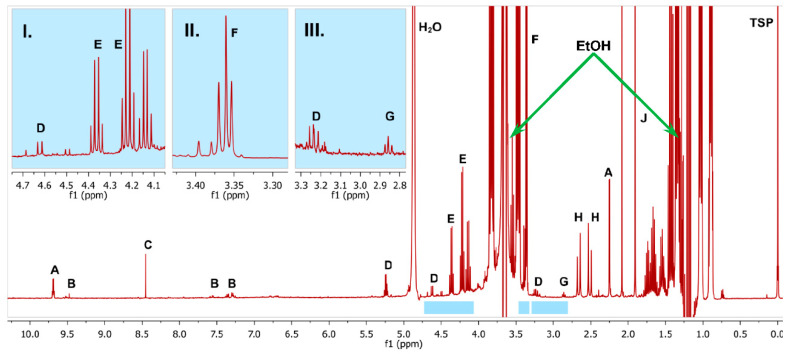
One-dimensional ^1^H NMR Spectrum of whisky spiked with several of the analytes. **I**, **II**, **III**: enlargements, see the blue bars under the main spectrum for the regions. The labelled signals are: A: acetaldehyde, B: HMF (positive, < LOQ: limit of quantification), C: formic acid, D: glucose, E: ethyl lactate, F: methanol, G: 2-phenylethanol (positive, < LOQ), H: citric acid, J: acetic acid, H_2_O: water (suppressed), TSP: trimethyl silylpropionate-d4; EtOH: ethanol (suppressed).

**Figure 5 foods-09-01355-f005:**
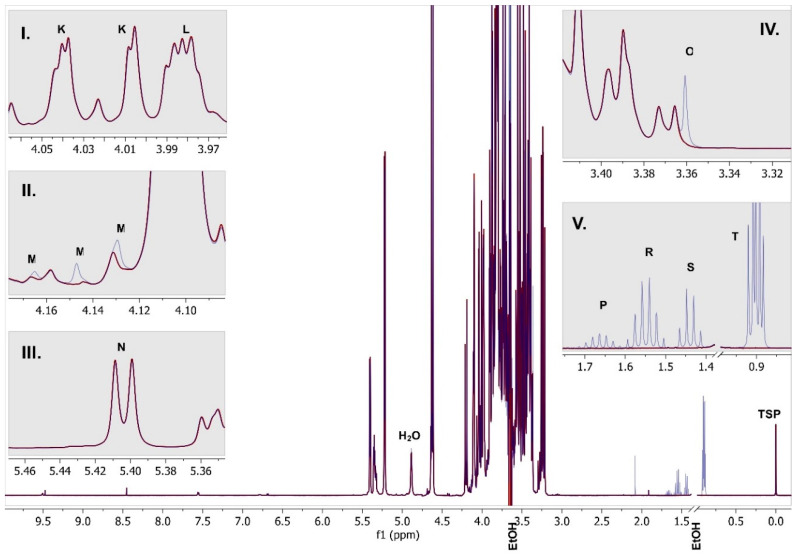
One-dimensional ^1^H NMR spectrum of herbal liqueur overlayed with spiked sample (blue). **I**.–**V**.: enlargements. The labelled signals are: K: fructose (dd), L: fructose (“pe”), M: ethyl acetate (q), N: sucrose (d), O: methanol, P: isopentanol (no), R: 1-propanol (hx), S: isopentanol (q), T: overlayed triplet of 1-propanol and doublet of isopentanol, H_2_O: water (suppressed), TSP: trimethylsilyl propionate-d4; EtOH: ethanol (suppressed, cut out).

**Table 1 foods-09-01355-t001:** Evaluated analytes in spirit drinks.

No.	Substance	Legal Limit(mg/L)	Relevant Working Range (mg/L)
1	1-Propanol	*	200–5000
2	2-Phenylethanol	*	30–2000
3	Acetaldehyde	*	10–1000
4	Acetic acid	*	10–5000
5	Citric acid	*	100–20,000
6	Ethyl acetate	*	20–10,000
7	Ethyl lactate	*	500–2000
8	Formic acid	*	10–5000
9	Fructose	*	200–150,000
10	Glucose	*	200–150,000
11	HMF		100–1000
12	Isobutanol	*	200–5000
13	Isopentanol	*	200–5000
14	Methanol	40 ^1^	
15	Sucrose	*	200–150,000

^1^ (for 40% strength vodka), * while no legal limit for the substance exists, there are group limits for some spirit drinks requiring a minimum of volatiles, and a maximum of acidity, esters, and sugar content (see e. g. annex II of [1] or [22]). HMF: 5-(hydroxymethyl)furfural.

**Table 2 foods-09-01355-t002:** Reference substances used in the QuantRef and the quality assurance solution.

No.	Substance	Signal Centre (ppm)	Fit & Integration Region (ppm)	Multiplicity	Coupling Constants *J* (Hz)	*N* _H_
1	Benzoate-Na ^1^	7.564	7.610 to 7.525	tt	7.34 and 1.72	1
2	Mannitol (1st) ^1^	3.878	3.920 to 3.840	dd	11.68 and 2.58	2
3	Mannitol (2nd) ^1^	3.690	3.730 to 3.640	dd	11.68 and 5.96	2
4	Propionate (1st) ^1^	1.060	1.120 to 1.000	t	7.66	3
5	Propionate (2nd) ^1^	2.187	2.250 to 2.120	q	7.66	2
6	Succinic acid ^2^	2.409	2.500 to 2.300	s	-/-	4

^1^ In both solutions. ^2^ Ingredient of the quality assurance solution only.

**Table 3 foods-09-01355-t003:** Characteristic NMR signals of the analytes in spirit drinks.

No.	Substance	Search Range (ppm)	Multiplicity	Coupling Constants *J* (Hz)	*N* _H_
1	1-Propanol	1.58–1.52	Hx	7.2	2
		0.92–0.89	T	7.38	3
2	2-Phenylethanol	2.90–2.80	T	6.92	2
3	Acetaldehyde	9.72–9.68	Q	2.94	1
		2.27–2.24	D	2.94	3
4	Acetic acid	1.925–1.9	S	–	3
5	Citric acid	2.72–2.65	D	15.05	2
		2.62–2.49	D	15.05	2
6	Ethyl acetate	4.18–4.12	Q	7.2	2
		2.10–2.07	S	–	3
7	Ethyl lactate	4.40–4.32	Q	6.95	1
		4.26–4.18	Q	7.13	2
8	Formic acid	8.48–8.44	S	–	1
9	Fructose ^1^	4.08–3.99	Dd	12.6 and 1.35	1
		4.01–3.95	Pe	1.69	1
10	Glucose ^2^	5.25–5.20	D	3.76	1
		4.66–4.58	D	7.96	1
		3.26–3.21	Dd	8.60 and 0.68	1
11	HMF	9.50–9.45	S	–	1
		7.60–7.50	D	3.85	1
		6.75–6.65	D	3.85	1
12	Isobutanol	3.375–3.35	D	6.62	2
		1.77–1.71	No	6.7	1
		0.89–0.75	D	6.74	6
13	Isopentanol	1.70–1.62	No	6.74	1
		1.455–1.41	Q	6.8	2
		0.915–0.895	D	6.68	6
14	Methanol	3.375–3.35	S	–	3
15	Sucrose	5.46–5.38	D	3.85	1
		4.25–4.18	D	8.72	1
		4.10–3.99	T	8.47	1

^1^ In aqueous solution fructose exists in five tautomeric forms [32]. Their different signals are partially discernible in the NMR spectrum and the intensities reflect the ratio between tautomers. Thus, correction factors were empirically determined by evaluating fructose spectra at different concentrations: 1.49 for the double doublet and 1.51 for the pentet. ^2^ In aqueous solution glucose exists in different anomeric forms. To calculate the glucose concentration, signals 1 and 2 (originating from the α and β anomeric protons respectively) were summed. Signal 3 originates from the C-2 proton of the β anomer only, a correction factor of 1.47 was determined (as for fructose).

**Table 4 foods-09-01355-t004:** Analytical limits for the validated analytes (in mg/L). LOD: limit of detection.

No.	Substance	LOD (mg/L)	LOQ (mg/L)	Range Evaluated (mg/L)
1	1-Propanol	11	26	10–509
2	2-Phenylethanol	8	19	24–243
3	Acetaldehyde	4	10	5–96
4	Acetic acid	3	6	1–229
5	Citric acid	3	8	10–100
6	Ethyl acetate	4	9	10–249
7	Ethyl lactate	11	27	11–546
8	Formic acid	3	7	10–100
9	Fructose	56	132	75–749
10	Glucose	35	84	50–1001
11	HMF	23	57	23–2333
12	Isobutanol	25	61	92–920
13	Isopentanol	74	173	51–505
14	Methanol	2	5	2–110
15	Sucrose	41	98	50–1002

**Table 5 foods-09-01355-t005:** Measurement uncertainties of analytes in spirit drinks.

No.	Substance	Offset *β*(0) (mg/L)	Measurement Uncertainty ±Δ*β* (mg/L)	Concentration Range for Calculation (mg/L)
1	1-Propanol	17	*β_found_* × 8% + 17	10–509
2	2-Phenylethanol	11	*β_found_* × 8% + 11	48–486
3	Acetaldehyde	10	*β_found_* × 8% + 10	5–964
4	Acetic acid	4	*β_found_* × 8% + 4	0.5–229
5	Citric acid	10	*β_found_* × 8% + 10	10–498
6	Ethyl acetate	15	*β_found_* × 8% + 15	10–497
7	Ethyl lactate	30	*β_found_* × 8% + 30	11–1090
8	Formic acid	6	*β_found_* × 8% + 6	10–1000
9	Fructose	200	*β_found_* × 8% + 200	75–7491
10	Glucose	62	*β_found_* × 8% + 62	50–1001
11	HMF	10	*β_found_* × 8% + 10	23–4667
12	Isobutanol	30	*β_found_* × 8% + 30	92–920
13	Isopentanol	90	*β_found_* × 8% + 90	51–1010
14	Methanol	4	*β_found_* × 8% + 4	2–110
15	Sucrose	53	*β_found_* × 8% + 53	50–1002

**Table 6 foods-09-01355-t006:** Stability of individual ingredients in spirit drinks.

No.	Deviation			
	Substance	After 1st Day (%)	After 2nd Day (%)	After 3rd Day (%)
1	1-Propanol	3.56 mg/L (0.6%)	4.47 mg/L (3.8%)	
2	2-Phenylethanol	---	---	---
3	Acetaldehyde	2.47 mg/L (10.2%)	1.17 mg/L (1.4%)	0.06 mg/L (1%)
4	Acetic acid	0.57 mg/L (5.1%)	0.33 mg/L (2.4%)	1.87 mg/L (6.7%)
5	Citric acid	−0.35 mg/L (0.4%)		
6	Ethyl acetate	0.63 mg/L (4.8%)	−15.2 mg/L (1.2%)	−0.8 mg/L (0.8%)
7	Ethyl lactate	−5.97 mg/ (2.0%)		
8	Formic acid	5.87 mg/L (15.7%)	2.9 mg/L (6.7%)	0.57 mg/L (3.5%)
9	Fructose	−94.0 mg/L (0.4%)	−63.5 mg/L (0.1%)	−0.5 mg/L (1.5%)
10	Glucose	17.1 mg/L (0.1%)	−177 mg/L (0.3%)	−1.0 mg/L (1.4%)
11	HMF	−1 mg/L (0.8%)	−1 mg/L (0.8%)	
12	Isobutanol	−2.5 mg/L (2.5%)	1.2 mg/L (0.6%)	
13	Isopentanol	−6.7 mg/L (2.1%)	−8.37 mg/L (1.5%)	
14	Methanol	0.23 mg/L (1.4%)	1 mg/L (0.1%)	−0.17 mg/L (3.2%)
15	Sucrose	−22.0 mg/L (0.1%)	−46.2 mg/L (0.2%)	1.6 mg/L (0.4%)

**Table 7 foods-09-01355-t007:** z-scores of the 2018 proficiency test “sprits drink analysis”.

No.	Substance	z-Score	No.	Substance	z-Score
1	1-Propanol	0.7	9	Fructose	–
2	2-Phenylethanol	–	10	Glucose	–
3	Acetaldehyde	−0.5	11	HMF	–
4	Acetic acid	–	12	Isobutanol	3.6
5	Citric acid	–	13	Isopentanol	1.3
6	Ethyl acetate	−1.4	14	Methanol	0.4
7	Ethyl lactate	0.7	15	Sucrose	–
8	Formic acid	–

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
