# Peer review of "Application of 1H Nuclear Magnetic Resonance Spectroscopy as Spirit Drinks Screener for Quality and Authenticity Control"

_foods, 2020, doi:10.3390/foods9101355_

Round 1

Reviewer 1 Report

In general, the paper entitled: “Application of 1H Nuclear Magnetic Resonance Spectroscopy as Spirits Screener for Quality and Authenticity Control”, describes experiments of a good quality with relevant analyses. The authors presented a new method for the quantitative analysis of compounds that may be present in alcoholic beverages. Currently, there are many analytical methods that can be used to analyze volatile compounds as well as organic acids and sugars in alcoholic beverages. What is new, however, is the development of a new method that allows the analysis of such chemically diverse compounds using a single analysis. The reported study is worthy of investigation, and might be very interesting for the research in this field. In addition, the authors described the possibility of practical application of the presented research.

However, here are few points which must be considered during revision of the manuscript:

#1 According to the EU directive, alcoholic beverages containing a minimum of 15% v/v ethanol are “spirits drinks”. Therefore, please use the phrase "spirits drinks" throughout the manuscript instead of "spirits" because "spirits" contains at least 88% v/v ethyl alcohol. Incorrect use of nomenclature may mislead the reader.

#2 In the introduction, please give examples of "spirits drink" and the chemicals they contain (in general).

#3 Please indicate which analytical methods are used to analyze individual "spirits drinks" depending on the concentration of ethyl alcohol and chemical compounds.

#4 Please avoid the term "spirits" for beverages containing carbohydrates, as distilled solutions which include "spirits" must not contain sugar.

#5 Please present the costs of the described method (for one sample) and compare it with the currently used analytical methods. It seems interesting for potential users.

#6 About 50 different chemical compounds (carbonyl compounds, esters of higher fatty acids, amyl alcohols, pyrazines, glycols, etc.) can be present in alcoholic beverages and raw spirit. Please describe whether the proposed method can be used for the simultaneous analysis of a larger group of organic compounds than just those indicated in the manuscript.

#7 Is the presented method suitable for qualitative and quantitative analysis of volatile by-products present in raw spirit?

Author Response

#1 According to the EU directive, alcoholic beverages containing a minimum of 15% v/v ethanol are “spirits drinks”. Therefore, please use the phrase "spirits drinks" throughout the manuscript instead of "spirits" because "spirits" contains at least 88% v/v ethyl alcohol. Incorrect use of nomenclature may mislead the reader.

Thank you for pointing out the nomenclature problems regarding “spirits”. Previous EU regulations such as Regulation 2870/2000 (Ref. #6) were arbitrary in their usage and used “spirits drinks” as well as “spirit drinks” throughout the text without any preference. However, the most recent EU regulation on spirit drinks (Regulation 2019/787, Ref. #1) changed the nomenclature to “spirit drinks”. Our text already corresponds to this most recent version of EU legislation (i.e. we intentionally use “spirit drinks” instead of “spirits drinks” throughout). (For details see: https://eur-lex.europa.eu/legal-content/EN/TXT/PDF/?uri=CELEX:32019R0787&from=EN).

It is also incorrect that spirit drinks must contain at least 88% v/v ethyl alcohol. Regulation 2019/787 states (Article 2): a spirit drink is an alcoholic beverage which complies with the following requirements: (a) it is intended for human consumption; (b) it possesses particular organoleptic qualities; (c) it has a minimum alcoholic strength by volume of 15 %, except in the case of spirit drinks that comply with the requirements of category 39 of Annex I.

While typically used synonymously, we have changed “spirits” to “spirit drinks” throughout to obtain better clarity.

#2 In the introduction, please give examples of "spirits drink" and the chemicals they contain (in general).

See minor addenda from l. 44 onwards.

#3 Please indicate which analytical methods are used to analyze individual "spirits drinks" depending on the concentration of ethyl alcohol and chemical compounds.

The desired information can be found from line 44 onwards.

#4 Please avoid the term "spirits" for beverages containing carbohydrates, as distilled solutions which include "spirits" must not contain sugar.

According to Regulation 2019/787, spirit drinks may contain carbohydrates either to round-off taste, e.g. in the case of rum, wine spirit or fruit spirits, or even higher amounts for sweetening such as in flavoured vodka, pastis or liqueurs.

#5 Please present the costs of the described method (for one sample) and compare it with the currently used analytical methods. It seems interesting for potential users.

Added to the text: Taking into account the spectrometer’s cost of purchase and annual operating costs, personnel costs and costs for consumables (all 2020 values), a pessimistic calculation of total costs per sample analysis adds up to approx. 50 EUR (60 USD). This amount factors in lower than 100% capacity utilization (due to R+D work, maintenance downtime etc.) For high throughput routine NMR experiments resulting in near full utilization, the costs per sample can be distinctly lower. Considering that one NMR experiment will yield quantitative data for several compounds in a sample, the application of NMR spectroscopy is comparably cheap.

#6 About 50 different chemical compounds (carbonyl compounds, esters of higher fatty acids, amyl alcohols, pyrazines, glycols, etc.) can be present in alcoholic beverages and raw spirit. Please describe whether the proposed method can be used for the simultaneous analysis of a larger group of organic compounds than just those indicated in the manuscript.

Added to the text: This method offers the potential to simultaneously quantify even more substances typically present in spirit drinks (or other foodstuffs): Any substance containing NMR-active nuclides, e.g. 1H, 13C, 19F…) will give rise to one or more NMR resonance signals. The pattern of these signals (their chemical shifts and multiplet splittings) is characteristic for the substance, especially if the substance has NMR-active nuclei in several different chemical moieties. The relative intensity of these signals is quantitatively linked to the stoichiometric number of nuclei giving rise to this resonance. NMR has been proven as a metrological primary method of measurement. [Malz, 2005; ref.14] Thus, if the spectrometer’s sensitivity has been determined once with a certified 1H reference material, one can quantify any other proton-containing substance. However, some caveats must be considered:

One downside of NMR is its comparably low sensitivity: Nowadays, NMR spectrometers with 7 to 14 T field strength (resulting in proton Larmor frequencies from 300 MHz to 600 MHz) offer a good compromise between pricing, stability / reliability and a wide application range for routine quantitative analyses. At these magnet strengths, the LOD/LOQ lies in the 5 to 50 ppm mass concentration range. For small molecules (e.g. methanol) 5 mg of analyte per kilogram sample convert to 5 mg·kg–1 / 32 mg·mmol–1 = 0,15 mmol·kg–1. If a longer experiment time (to accumulate more FID scans) is acceptable or a cryo probe (having a considerably lower electronic noise level) can be used or works with a magnet of higher field strength, the LODs/LOQs will be significantly lower.

Analysing very complex mixtures, another problem can be the “crowding” of signals, especially in the chemical shift range around 3.3 to 4 ppm, where many carbohydrates resonate. The unambiguous assignation of signals to their molecules is tough and often only possible using a 2D spectrum such as the JRES. If signals from different nuclei overlap, they need to be deconvoluted prior to quantification, using complex curve fitting routines, valid knowledge of coupling constants and the relative signal heights in multiplets.

#7 Is the presented method suitable for qualitative and quantitative analysis of volatile by-products present in raw spirit?

Added to the text: Besides common commercial spirit drink samples, our lab investigated several dozens of samples of raw spirit (drinks) and “moonshine” (esp. eastern European “Samogon”). The method is already validated for several volatiles (fusel oils) relevant in raw spirits.

Reviewer 2 Report

In this work, the authors propose a methodology for detecting and quantify the main ingredients present in spirits. This methodology is based on the use of NMR spectroscopy assisted by an implemented automatic routine that performs the various steps. With this methodology, the authors are able to simultaneously quantify 15 ingredients present in the spirits analyzed. The subject of the work is very interesting and promising if it is true that it satisfies, as the authors say, the requirements of the  EU current regulations. Only a few things should be clarified.

In particular the authors should consider the comments below.

  • In the section 2.3. “ Proton NMR experiments”, the authors report all the acquisition parameters for the three experiments they performed (zgpr, noesy and jres). It seems to me a little too redundant, as the authors themselves say in the course of the work, these are commonly used pulse sequences.
  • In the section 3.1. “ Relevant analytes……”, line 442, what is meant by “ real spirits samples”?

Author Response

In the section 2.3. “ Proton NMR experiments”, the authors report all the acquisition parameters for the three experiments they performed (zgpr, noesy and jres). It seems to me a little too redundant, as the authors themselves say in the course of the work, these are commonly used pulse sequences.

The basic pulse experiments are routine, but the detailed acquisition parameters are the ones used in our validation (and routine analyses). Different parameter settings may yield good results but could also produce faulty data, thus need to be validated again.

In the section 3.1. “ Relevant analytes……”, line 442, what is meant by “ real spirits samples”?

Real in this context means “commercial off the shelf”.